# Bad-metal relaxation dynamics in a Fermi lattice gas

W. Xu[1], W.R. McGehee [2], W.N. Morong [3] & B. DeMarco[3]

Electrical current in conventional metals is carried by electrons that retain their individual character. Bad metals, such as the normal state of some high-temperature superconductors, violate this scenario, and the complete picture for their behavior remains unresolved. Here, we report phenomena consistent with bad-metal behaviour in an optical-lattice Hubbard model by measuring the transport lifetime for a mass current excited by stimulated Raman transitions. We demonstrate incompatibility with weak-scattering theory and key characteristics of bad metals: anomalous resistivity scaling consistent with $T$-linear behavior, the onset of incoherent transport, and the approach to the Mott–Ioffe–Regel limit. Our work demonstrates a direct method for determining the transport lifetime, which is critical to theory but difficult to measure in materials, and exposes minimal ingredients for bad-metal behavior.

[1] Department of Physics, Massachusetts Institute of Technology, Cambridge, MA 02139, USA. [2] Center for Nanoscale Science and Technology, National Institute of Standards and Technology, Gaithersburg, MD 20899, USA. [3] Department of Physics, University of Illinois at Urbana-Champaign, Urbana, IL 61801, USA. Correspondence and requests for materials should be addressed to B.D. (email: bdemarco@illinois.edu)

Landau's Fermi liquid theory successfully describes the behavior of interacting fermionic particles for a wide range of materials, such as electrons in simple metals and liquid helium-3[1]. In some metals, Fermi liquid theory fails when strong correlations or fluctuations are present[2]. Also known as bad, or strange, metals, these states present anomalous properties such as resistance that does not follow the Fermi liquid prediction $T^2$, sometimes scaling as $T$ instead or exhibiting more complex phenomena[2,3]. The resistivity of bad metals also does not saturate[4,5] as temperature is increased into the regime where the Mott–Ioffe–Regel (MIR) limit is violated and the apparent mean-free-path is shorter than the interatomic spacing[4,6]. This lack of saturation implies that quasiparticles are absent (e.g., refs. [7–9]. for an overview), as does the lack of particle-like excitations in photoemission spectroscopy[10].

Understanding the origin of bad-metal behavior is a key problem in condensed matter physics, which may be important to resolving questions related to high-temperature superconductivity[11] and Mott quantum criticality[12]. This problem has been studied using varied theoretical frameworks, including anti-de Sitter space-conformal field theory (AdS-CFT) holographic duality[8,9,13], the Sachdev–Ye–Kitaev model[14], high-temperature series expansions[15], and dynamical mean-field theory (DMFT) (e.g., refs. [16,17]). DMFT, in particular, has shown $T$-linear resistivity that can exceed the MIR limit and a regime below the MIR limit in which quasiparticles persist[18]. Despite this extensive work, a full picture for bad-metal behavior is incomplete. For example, there is evidence that electron–phonon interactions play an important role[19–22]. The many scattering mechanisms present in solids, such as disorder, phonons, and interactions between quasi-particles, scale differently with temperature, which complicates efforts to obtain a complete understanding.

Ultracold fermionic atoms trapped in optical lattices, which realize the Fermi-Hubbard model[23–25], provide a well-controlled platform free of phonons and impurities with well controlled and understood microscopic parameters to study bad metal phenomenology[15]. In ultracold gas experiments with fermionic atoms, photoemission spectroscopy has been used to probe the spectral function in the Bose–Einstein condensate (BEC)–Bardeen–Cooper–Schrieffer (BCS) crossover for a trapped gas, and a failure of Fermi liquid theory was discovered[26].

Transport measurements such as diffusion in a 2D lattice gas[27], shear viscosity in a unitary Fermi gas[28], and spin diffusion[29] have also explored the effect of strong interactions on various relaxation processes. In this paper, we describe a method for measuring the decay rate of a mass current and inferring the analog of electrical resistivity for a two-component fermionic gas composed of $^{40}K$ atoms trapped in a cubic optical lattice. A net current consisting of a flow of spin-polarized atoms shifted in quasimomentum is created using stimulated Raman transitions (Fig. 1). By fully resolving the decay dynamics of the current, we deduce the transport lifetime induced by collisions with atoms in the other spin state. The analog of resistivity is inferred from the transport lifetime and the atomic density.

## Results

**Mass current generation.** We prepare the gas in a metallic state by slowly superimposing the optical lattice after cooling in an optical dipole trap to temperatures $T \approx 0.2$–$1.2\,T_F$, where $E_F = k_B T_F$ is the Fermi energy (see Methods). The temperature of the gas is sufficiently low for the atoms to realize a single-band Hubbard model described by the Hamiltonian

$$H = -t \sum_{\langle i,j \rangle, \sigma} \left( \hat{c}_{j\sigma}^{\dagger} \hat{c}_{i\sigma} + h.c. \right) + U \sum_i n_{i,\downarrow} n_{i,\uparrow} + \sum_{i,\sigma} m\bar{\omega}^2 r_i^2 n_{i,\sigma}/2,$$

where $i$ indexes the lattice sites, $\langle \rangle$ indicates a sum over neighboring sites, $\sigma = \uparrow, \downarrow$ indexes spin, $\bar{\omega}$ is the geometric mean of the dipole trap frequencies, $r_i$ is the distance from site $i$ to the trap center, $\hat{c}_i^{\dagger}$ ($\hat{c}_i$) creates (annihilates) an atom from site $i$, $n_{i,\sigma} = \hat{c}_i^{\dagger} \hat{c}_i$ is the number operator, $t$ is the Hubbard tunneling energy, and $U$ is the on-site Hubbard interaction energy[23]. Two hyperfine states $|F = 9/2, m_F = 9/2\rangle$ ($|\uparrow\rangle$) and $|F = 9/2, m_F = 7/2\rangle$ ($|\downarrow\rangle$) play the role of the electron spin. The metallic regime is achieved by tuning the number of atoms $N$ so that $E_F \approx 6t$ (corresponding to 0.5 particles of each spin per site in the center of the lattice at $T = 0$) and the lattice potential depth $s$ to sample $U/t \approx 2.3$–9.0. To create a well-characterized initial state, the gas is spin polarized by removing the $|\downarrow\rangle$ atoms before turning on the lattice. We use exact eigenstates[30,31] and measurements of $N$ and $T$ to estimate an effective chemical potential $\tilde{\mu}$ and temperature $\tilde{T}$ of the initial metallic lattice gas (see Supplementary Note 1).

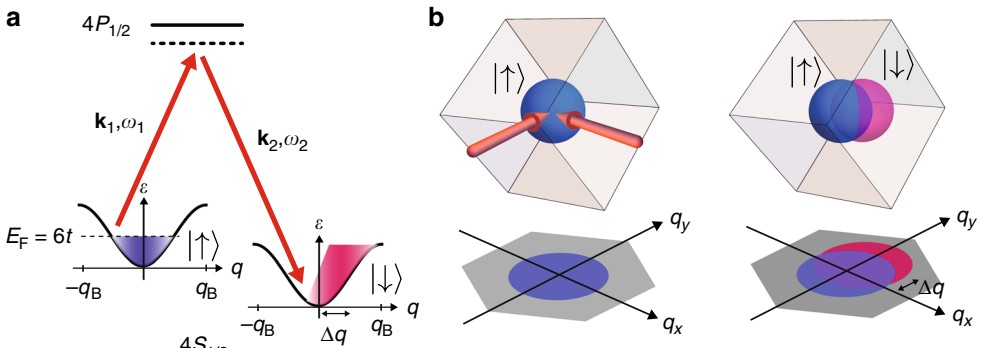

**Fig. 1** Schematic diagram of method used to generate current via Raman transition. **a** A spin-polarized $|\uparrow\rangle$ (blue) gas is prepared in a metallic state in the ground band (with dispersion $\varepsilon$) of the lattice. A pair of Raman beams (red arrows) with frequencies $\omega_1$, $\omega_2$ and wavevectors $\mathbf{k}_1$, $\mathbf{k}_2$ are used to quickly transfer atoms from the $|\uparrow\rangle$ to the $|\downarrow\rangle$ (red) state via the $4S_{1/2} \rightarrow 4P_{1/2}$ electronic transition. The frequency difference $\delta\omega = \omega_1 - \omega_2$ is tuned to be resonant, and the Raman pulse equally samples all quasimomenta $q$ in the Brillouin zone, which ranges along one direction from $-q_B$ to $q_B = \hbar\pi/d$, where $d \approx 390$ nm is the lattice spacing. The Raman transition introduces a momentum shift $\Delta q$ for the atoms in the $|\downarrow\rangle$ state, resulting in a net current of $|\downarrow\rangle$ atoms. **b** The quasimomentum distribution in 3D Brillouin zone is shown before (left) and after (right) the Raman pulse. The initial quasimomentum shift has approximate magnitude of 0.5 $q_B$ and is aligned with the direction of $q_y$ and $\delta\mathbf{k}$, which is along the $[-1, -1, -1]$ direction. The imaging procedure projects the quasimomentum distributions along the $[1, -\sqrt{2}, 1]$ direction, so that the Brillouin zone has a hexagonal shape in the imaging plane, which has axes $q_x$ and $q_y$

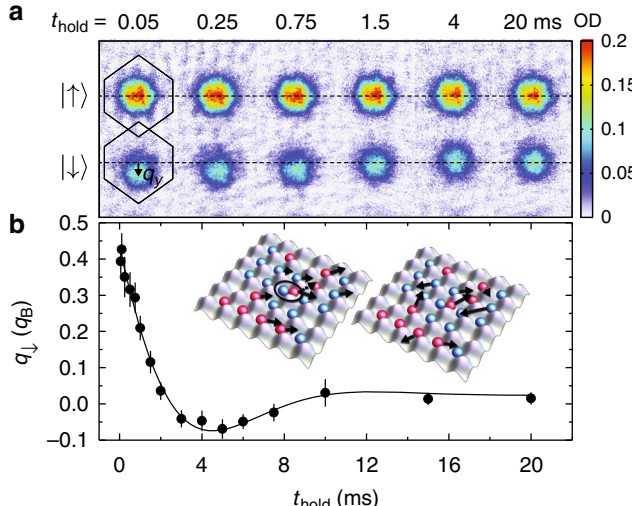

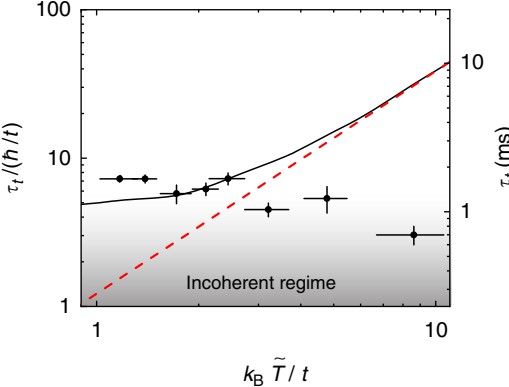

**Fig. 3** Transport lifetime at varied temperature and fixed interaction strength $U/t = 2.3$. A logarithmic scale is used to display the data. The lifetime $\tau_t$ is shown in units of the tunneling time (left axis) and in ms (right axis). The slope from a linear fit to the data (not shown) has a negative slope at greater than a 99.5% confidence level. The measurements are compared with a FGR weak-scattering calculation (black solid line) and a scaling prediction (red dashed line, which was fixed to match the FGR result at the highest temperature point) in the Maxwell-Boltzmann limit. For the FGR calculation, $N$ is fixed to the average value from the data. The vertical error bars show the uncertainty in the fit used to determine $\tau_t$, and the horizontal error bars display the uncertainty in $T$ from time-of-flight thermometry

**Fig. 2** Measurement of transport lifetime. **a** Quasimomentum distribution for the $|\uparrow\rangle$ and $|\downarrow\rangle$ components for different evolution times after the Raman pulse for $T/T_F = 0.23$ and $U/t = 2.3$ ($s = 4E_R$). The dashed lines mark $q = 0$, and the hexagons are the first Brillouin zone (BZ) projected onto the imaging plane. The color bar shows the measured optical depth (OD). Fits to these quasimomentum profiles are used to determine the average momentum $q_\downarrow$ of the $|\downarrow\rangle$ component (see Methods). **b** The insets shows how momentum-changing collisions between atoms in different spin states relax the initial current (left) so that the net quasimomentum vanishes at long times (right). The momentum $q_\downarrow$ is fit (solid line) to a solution of the Boltzmann equation to determine the transport lifetime. Each point is the average of 5–10 measurements, and the error bars show the standard error of the mean

A current consisting of approximately 30% of the atoms transferred to the $|\downarrow\rangle$ state and shifted in quasimomentum is created using a pulse of two laser beams focused onto the gas (Fig. 1). Based on semi-classical, non-interacting thermodynamics, we estimate that the Raman excitation increases the total energy of the gas by less than 10%. The quasimomentum profiles of the atoms in the $|\uparrow\rangle$ and $|\downarrow\rangle$ states are separately imaged after evolution time $t_{hold}$ in the lattice using bandmapping[31] and spin-resolved time-of-flight imaging (see Methods). Sample images for $s = 4E_R$, corresponding to $U/t = 2.3$, and $T/T_F = 0.23$ before turning on the lattice ($k_B \tilde{T} = 1.3t$) are shown in Fig. 2a. The quasimomentum distribution of the $|\uparrow\rangle$ gas is unaffected by the Raman pulse, while the $|\downarrow\rangle$ gas is displaced along the wavevector difference $\delta\mathbf{k} = \mathbf{k}_1 - \mathbf{k}_2$ between the Raman beams. The $|\downarrow\rangle$ atoms, therefore, form a net current proportional to their average quasimomentum $q_\downarrow$ (Fig. 2b), which is determined by fitting the images to a Gaussian function (see Methods).

**Transport lifetime**. The decay of the current caused by momentum-changing collisions between atoms in $|\downarrow\rangle$ and $|\uparrow\rangle$ states is apparent for different evolution times in the lattice following the Raman pulse (Fig. 2b). The characteristic decay time, which is the transport lifetime $\tau_t$, in our experiments is a few milliseconds. Effects besides interactions play a minor role in the excitation dynamics. The relaxation is too fast for the trap oscillations to play a significant part in the dynamics or for the motion of the $|\uparrow\rangle$ atoms to be affected for most of the temperatures and interaction strengths we sample. We have also checked that dephasing of atomic trajectories with different initial quasimomenta and trap anharmonicity do not significantly contribute to the relaxation via classical dynamics simulations and measurements employing spin-polarized gases (see Methods).

The Boltzmann formalism provides an intuitive picture to relate the transport lifetime with microscopic scattering processes and resistivity[1]. In solids, $\tau_t$ is usually inferred from resistivity, but here we measure it directly. The decay rate of $q_\downarrow$ is the inverse of the transport lifetime $\tau_t$ averaged over the density profile (see Methods). To determine $\tau_t$, data such as those shown in Fig. 2b are therefore fit to a model based on the Boltzmann equation (see Methods), which is similar to an approach that has been used to determine collision cross sections[32] and observe Pauli blocking[33,34] in weakly interacting Fermi gases.

Measurements of $\tau_t$ for different temperatures at fixed $U/t = 2.3$ (corresponding to $s = 4 E_R$) are shown in Fig. 3. We compare these data to the thermal-limit scaling prediction for scattering between trapped quasiparticles and a Fermi's golden rule (FGR) calculation that accounts for collisions between atoms in quasimomentum states (see Supplementary Note 2). In the Maxwell-Boltzmann limit ($T > T_F$), the scattering time between quasiparticles scales as $1/\langle v \rangle n_{dwd} \propto T^{3/2}$, where $\langle v \rangle$ and $n_{dwd}$ are the thermally averaged speed and density-weighted (thermally averaged) density (see Methods). The more sophisticated FGR calculation, which has no free parameters, agrees with this behavior at high temperature. A FGR approach has been used to accurately calculate relaxation times for trapped gases in the weakly interacting regime[33], but may be expected to fail for the strong interactions ($U \gtrsim 2t$) sampled by our measurements. Our FGR calculation has no free parameters, fully accounts for the trap and quantum statistics, and averages over a thermodynamic distribution of quasimomenta based on the inferred $\tilde{T}$ and $\tilde{\mu}$. Based on general principles, this approach predicts that $\tau_t$ decreases for stronger interactions, since the rate of scattering events increases, and that $\tau_t$ increases at higher temperatures, because the density is reduced. Furthermore, while Pauli blocking limits the phase space for scattering and causes the equilibrium collision rate to vanish at zero temperature[33], $\tau_t$ for our measurement remains finite at $T = 0$ because the Raman excitation creates unfilled quantum states.

The temperature dependence of $\tau_t$ shown in Fig. 3 shows a trend strikingly opposite to that predicted by scattering theory: at greater than a 99.5% confidence level, the transport lifetime decreases for higher temperatures. While this behavior is standard in solid metals, it is surprising for trapped gas in this temperature regime—a quantity proportional to the mean time between collisions such as $\tau_t$ is expected to increase at higher temperatures because the density decreases as the gas expands into a larger volume of trap. For these data, we vary the temperature of the gas before turning on the lattice from $T/T_F \approx$ 0.2–1.2, which leads to $k_B \tilde{T}/t \approx 1-8$. The upper end of this regime cannot be explored in solid metals, where $T_F = (1-15) \times 10^4$ K, which is well above the melting temperature. The measured transport lifetime agrees with weak scattering theory within 30% at the lowest temperatures. As the temperature is increased, $\tau_t$ decreases by approximately a factor of two, while the weak scattering calculation predicts that $\tau_t$ increases by a factor of 6, leading to a disagreement of over 60 standard errors at the highest temperature. This discrepancy cannot be explained by an error in density—we have verified that the density of the gas decreases across this range and is consistent with thermodynamic calculations via in-situ imaging (see Supplementary Note 1).

**T-linear resistivity**. A higher-than-expected increase in scattering with temperature is characteristic of bad metals. The onset of another key signature of bad metals—incoherent transport—is also apparent in Fig. 3. Transport in a metal becomes incoherent[35] when the lifetime of states with well-defined momentum is comparable to the characteristic single-particle timescale, which is the tunneling time $\hbar/t$. This regime is approached at high temperatures (Fig. 3) and at high interaction strengths (see Methods). The commensurability of timescales signals the breakdown of a central assumption underlying Fermi liquid theory and the failure of the quasiparticle picture. The inverse dependence of $\tau_t$ on $T$ evident in Fig. 3 cannot be explained by any known quasiparticle theory, and therefore suggests that quasiparticles are absent.

To expose other bad-metal behaviors and compare with DMFT predictions, we infer a dimensionless resistivity $\rho$ from the measured $\tau_t$ and the relationship between the Kubo and Boltzmann formalisms (first addressed by Thouless in ref. [36] see also, for example, ref. [37]). In $\rho$, we account for the harmonic trap in our experiment, which introduces two features absent in bulk solids and DMFT: a spatially inhomogeneous density profile and a temperature-dependent density. To address these differences, we define the dimensionless analog to resistivity as $\rho = \left( \frac{\tau_t}{\hbar/t} n_{dwd} d^3 \right)^{-1}$ (see Methods). Here, the density-weighted-density $n_{dwd}$ (see Supplementary Note 4) is used to account for the average of scattering processes over the inhomogeneous density profile and is determined using semi-classical thermodynamics calculations based on $\tilde{\mu}$ and $\tilde{T}$. The transport lifetime is normalized to the tunneling time, which incorporates the dependence on the effective mass.

The dimensionless resistivity $\rho$ corresponding to the data in Fig. 3 and separate measurements of $\tau_t$ at fixed $T/T_F \approx 0.25$ and varied $U/t$ (tuned via $s$, see Methods) are shown in Fig. 4. The data fit well to scaling predictions for a bad metal from DMFT simulations of the Hubbard model[16,38] (see Supplementary Note 3). For our lattice parameters and regime of temperature, DMFT predicts that the resistivity scales quadratically with interaction strength and linearly with temperature. The $(U/t)^2$ scaling in Fig. 4a can be accommodated by Fermi liquid theory, while, in contrast, the scaling consistent with $T$-linear evident in

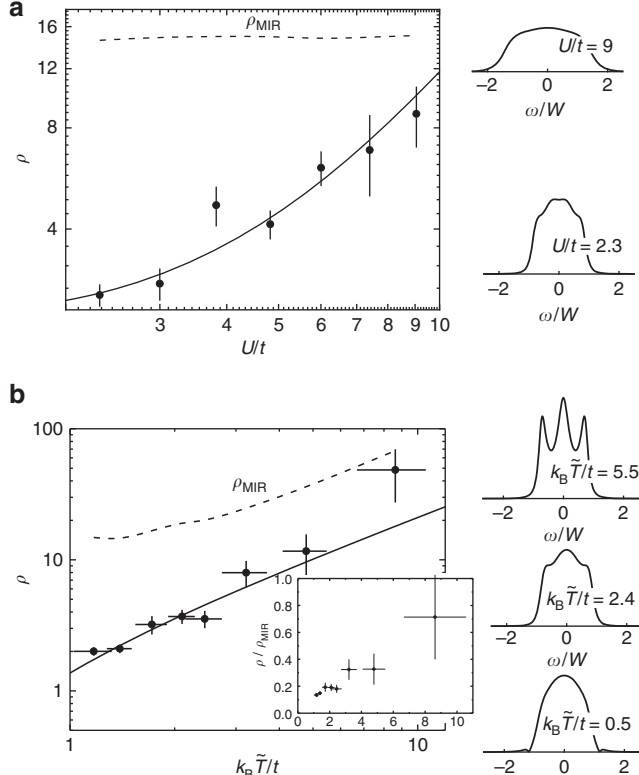

**Fig. 4** Interaction and temperature dependence of the dimensionless resistivity $\rho$. These data correspond to those shown in Figs. 3 and 5. **a** Interaction scaling, fit (solid line) fit to DMFT scaling law $U^2$ with reduced $\chi^2 = 0.99$. **b** Temperature scaling, fit (solid line) to DMFT scaling law $T$ with reduced $\chi^2 = 1.76$. The vertical error bars include the uncertainty in the fit used to determine $\tau_t$ and the uncertainty in $n_{dwd}$ from the measurements of $N$ and $T$. Orthogonal distance regression is used for fitting to accommodate the horizontal error bars in **b**. The plots along the right side of the figure show the local spectral function $A(\omega)$ using a constant vertical scale, and the energy $\omega$ relative to the Fermi energy is in units of the half bandwidth $W$. $A(\omega)$ is calculated at $U/t = 2.3$ for **b**. The inset in **b** shows the ratio between the dimensionless resistivity and the MIR limit

Fig. 4b is contradictory with Fermi liquid theory and is a signature of a bad metal. In normal solid metals, scattering is dominated by Pauli blocking, which leads to $T^2$ scaling of resistivity. Bad metals deviate from this behavior, either demonstrating $T$-linear scaling or more complex phenomena. The $T$-linear scaling evident in Fig. 4b is also inconsistent with weak scattering theory for a trapped gas. For trapped gases in the temperature range we explore and with the excitation present, Pauli blocking is suppressed, and the resistivity $\rho$ is therefore expected to be independent of temperature because the scattering time is inversely proportional to $n_{dwd}$ for fixed $N$, $t$, and $U$.

**Approaching the MIR limit**. The onset of another characteristic of bad metals is also evident in Fig. 4: continual growth toward the MIR limit as temperature is increased. The MIR limit defines the regime in which semiclassical transport theory is valid and current-carrying particles are a legitimate concept[4,6]. In solids, the MIR limit is $l \approx d$ [6], where $l$ is the mean-free path and $d$ is the atomic lattice spacing. This condition must be modified to $l \approx n^{-1/3}$ for optical lattices, which are free from impurities and phonons, and the only scattering is between particles with separation $n^{-1/3}$. Using this definition, we show the MIR-limited

resistivity $\rho_{MIR}$ in Fig. 4 determined from $\tau_{MIR} = \langle n_\uparrow^{-1/3}\rangle/\langle v_\downarrow\rangle$. The resistivity steadily escalates toward $\rho_{MIR}$ with interaction strength (Fig. 4a). The behavior in Fig. 4b is more subtle, since $\rho_{MIR}$ is a strong function of temperature. The ratio $\rho/\rho_{MIR}$ shown in the inset reveals that $\rho$ continuously approaches the MIR limit as the temperature of the gas is increased. While we measure a steady increase of resistivity with temperature and interaction strength, a violation of the MIR limit is not evident in our data. We cannot sample higher temperatures and interaction strengths, where a MIR violation may occur, because the mass current (which vanishes in the $T, U/t \to \infty$ limits) becomes too small to resolve.

## Discussion

One way to understand the effect of the strong interactions on the system is through the change in the local spectral function $A(\omega)$[16] (see Supplementary Note 3). This quantity captures changes in the density of states caused by interactions and temperature. At low temperature, $A(\omega)$ consists of a single band centered at the Fermi energy, which is broadened by interactions from the non-interacting bandwidth $2W$ (Fig. 4a). As the temperature is raised, spectral weight is redistributed to peaks centered at approximately $\pm U/2$ (Fig. 4b). This reduction in the spectral weight near the Fermi surface gives rise to a resistivity that depends on the changing nature of the quasiparticles in addition to their scattering rate, providing insight into the qualitative failure of the weak scattering calculation.

To our knowledge, our measurement of scaling consistent with $T$-linear samples the highest temperatures relative to the Fermi temperature, and, along with the concurrent work reported in ref. [39], is evidence for this behavior in an ultracold-gas Hubbard model. Because this system has precisely known microscopic parameters and is well isolated from the environment, our measurements provide direct evidence—consistent with the predictions from DMFT and other techniques (e.g., refs. [15,40])—that the minimal ingredients of strongly interacting lattice fermions contained in the Fermi-Hubbard Hamiltonian are sufficient to cause some characteristic bad-metal dynamics. In the future, rf spectroscopy measurements in this system may reveal information about $A(\omega)$ directly[26]. Furthermore, additional effects present in solids can be added in a controllable fashion. The influence of disorder can be investigated via, for example, applying optical speckle[41], and the impact of phonons could be explored using mixtures of different species[42].

## Methods

**Lattice gas and mass current preparation.** Ultracold gases composed of $^{40}$K atoms are cooled to temperatures below $T_F$ in a crossed-beam 1064 nm optical dipole trap using standard techniques. A 3 G static magnetic bias field is applied. The final trap depth during evaporative cooling is adjusted to control the temperature and atom number. After cooling, the optical trap depth is slowly increased to the same value for all the data presented in this paper, resulting in trap frequencies $(47.9 \pm 0.4)$, $(98 \pm 1)$, and $(114 \pm 2)$ Hz. In conjunction with a transiently applied magnetic field gradient, a microwave-frequency swept magnetic field is used to remove all atoms in $|\downarrow\rangle$ state (by transferring atoms to the $|F = 7/2, m_F = 7/2\rangle$ state) before the lattice beams are ramped on in 100 ms.

We apply a 25 μs-long Raman pulse after loading atoms into optical lattice. The pulse time is sufficiently short for atoms to be uniformly excited across the BZ. The pair of Raman beams is derived from a cavity-stabilized diode laser that is 40 GHz red-detuned from the $D1$ transition. The timing of the Raman pulses, pulse power, and $\delta\omega$ are controlled using acousto-optic modulators. The Raman transition generates atoms in a superposition of the $|\uparrow\rangle$ and $|\downarrow\rangle$ states, and the relative amplitude in each state depends on the initial quasimomentum. Subsequently, the coherence of the superposition decays after the pulse. The decoherence timescale measured using a Ramsey pulse sequence is approximately 0.08 ms, and therefore the Raman transition can be treated as an instantaneous process for the relaxation measurements.

**Transport lifetime measurement.** After holding atoms in the lattice for variable times following the Raman transition, we ramp down the lattice potential in 0.1 ms and release the gas from the trap. A magnetic field gradient is applied along the $y$ direction during time-of-flight expansion to spatially separate atoms in the $|\uparrow\rangle$ and $|\downarrow\rangle$ states. The images of each spin component are separately fit to a Gaussian distribution to obtain the center-of-mass (COM) position $y_\uparrow$ ($y_\downarrow$) for the $|\uparrow\rangle$ ($|\downarrow\rangle$) atoms. This COM position is translated into a net quasimomentum shift as $q_{\uparrow,\downarrow}/m = [y_{\uparrow,\downarrow} - y_{0,(\uparrow,\downarrow)}]/t_{TOF}$, where $t_{TOF}$ is the expansion time, and $y_{0,(\uparrow,\downarrow)}$ is the COM position without a Raman pulse. The $|\uparrow\rangle$ atoms remain nearly at rest for all of the data—the maximum $q_\uparrow$ is an order of magnitude smaller than $q_\downarrow$ after the Raman pulse. The net current is therefore proportional to $q_\downarrow$. The effect of the dipole force from the Raman beams is too small to affect our analysis.

We fit the time evolution of $q_\downarrow$ to a solution of the Boltzmann equation[1,43] $\frac{\partial}{\partial t} q_\downarrow = -m\Omega^2 y_\downarrow(t) - q_\downarrow/\tau_t$. The first term on the right-hand side of this equation accounts for the harmonic trap, where $m$ is the atomic mass, $y_\downarrow$ is the in-trap position relative to the trap center, and $\dot{y}_\downarrow = q_\downarrow/m$. These quantities should all be understood as thermodynamic averages. The damping force (that is, the second term on the right-hand side) is from collisions between atoms in different spin states. The transport lifetime is $\tau_t$ as reported in this work. The solution to this equation with initial conditions at $y_\downarrow = 0$ and $q_\downarrow = q_0$ gives

$$q_\downarrow = \frac{q_0}{2\sqrt{\phi(\tau_t)}}\left[\left(1 + \sqrt{\phi(\tau_t)}\right)e^{-\frac{t}{2\tau_t}\left(1 + \sqrt{\phi(\tau_t)}\right)} + \left(-1 + \sqrt{\phi(\tau_t)}\right)e^{-\frac{t}{2\tau_t}\left(1 - \sqrt{\phi(\tau_t)}\right)}\right].$$

(1)

Here, $\phi(\tau_t) = 1 - 4\tau_t^2\Omega^2$, and $\tau_t$, $\Omega$, $q_0$, and an offset are free parameters in the fit. Nonlinear effects are small—the data fit well to this model of linear dissipation with an adjusted $R^2 = 0.7$–$1.0$ for all the data used in this work and the fit residuals display no systematic effects (see Supplementary Note 5). An offset is necessary to account for drifts in the center of the equilibrium gas caused by changes in the dipole trap and stray magnetic field gradients. We find that this offset corresponds to less than a pixel in the images for all the data used in this work.

**Interaction dependence.** Independent measurements of how $\tau_t$ depends on interaction strength are used to infer $\rho$ for Fig. 4a. For these measurements, shown in Fig. 5, $U/t = 2.3$–$9$ is tuned at fixed $T/T_F \approx 0.25$ by changing the lattice potential depth from $s = 4$–$7 E_R$. As is the case at high temperature, the incoherent regime is achieved at high $U$. The data are compared with the FGR weak scattering calculation (solid line). With $E_F$ fixed to approximately $6t$, the weak scattering calculation predicts $\tau_t \propto t/U^2$. The measured $\tau_t$ normalized to the tunneling time $\hbar/t$ agrees within 10% with the weak scattering prediction at the lowest interaction strength. While $\tau_t$ follows the same trend as weak scattering theory, it does not decrease with increasing interaction strength as rapidly as the weak scattering prediction. Across the range we sample, the weak scattering theory predicts that $\tau_t$ decreases by a factor of 10 (see inset), while the measured value changes only by a factor of 2, leading to a disagreement of five standard errors at $s = 7 E_R$. This discrepancy may be explained by the change in $A(\omega)$ shown in Fig. 4. The FGR calculation assumes that $A(\omega)$ is a delta-function, while DMFT predicts a very broad peak at high $U/t$, which implies a higher scattering rate.

**Dephasing time for a spin-polarized gas.** We have assumed that the atoms have a free-particle dispersion in this approach in order to develop a simple, closed form

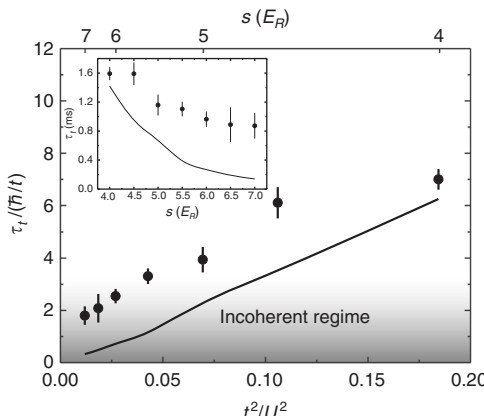

**Fig. 5** Transport lifetime normalized to the tunneling time at varied interaction strength. The inset shows the measured $\tau_t$ determined from fits to data such as those shown in Fig. 2b. For the FGR weak scattering calculation, $\tilde{T}$ and $\tilde{\mu}$ are fixed to the values for these parameters averaged across the data set. The vertical error bars show the uncertainty in the fit used to determine $\tau_t$

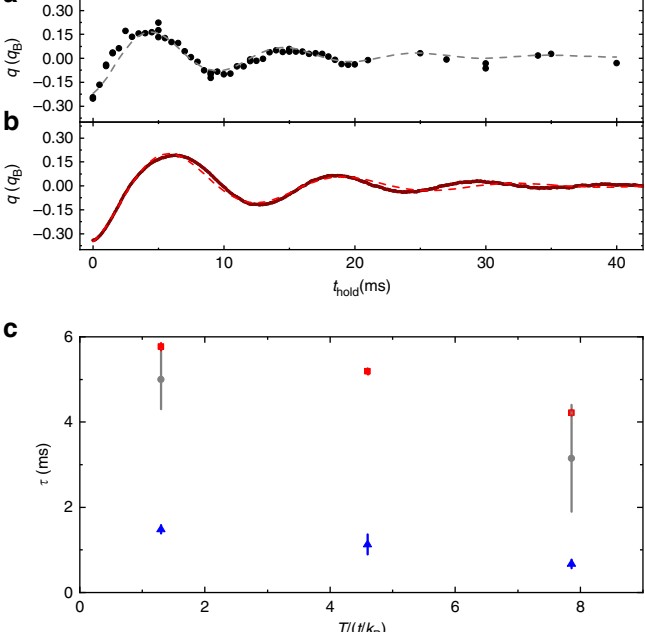

**Fig. 6** Dephasing of motion for a non-interacting gas in an $s = 4$ lattice. **a** Measured average quasimomentum $q$ along the impulse direction for a spin-polarized (non-interacting) gas. For these data, $N \approx 42,000$ and $T/T_F \approx 0.25$ in the trap before turning on the lattice. The line is a fit used to determine the dephasing time. **b** Simulated average quasimomentum $q$ along the impulse direction for a non-interacting gas. For this simulation, $\tilde{T} = 1.3(t/k_B)/3$ and $\tilde{\mu} = 4.5t/3$ in the lattice. The deviation of the fit at long times is a result of the anharmonicity of the lattice gas, which is not accounted for in the fit. **c** Comparison of dephasing and transport lifetime $\tau$ at different temperatures. The dephasing time measured using a spin-polarized gas are shown using red circles, and the measured transport lifetime for a spin-mixed gas is shown using blue triangles. The simulated dephasing time is plotted using gray squares. For the simulated time, the temperature is three times smaller than the value on the abscissa. The chemical potentials used in the simulation are $4.5t/3$, $-4.5t/3$, and $-18.2t/3$. The error bars represent the uncertainty in the fit used to determine $\tau$

fitting function for the data. Because of the tight-binding dispersion, the gas in the lattice is not harmonic, and hence Kohn's theorem does not hold. Therefore, center-of-mass motion such as we excite will decay as individual atomic trajectories dephase. We experimentally probe and theoretically model this dephasing time to verify that the relaxation we measure is dominated by interaction-induced scattering between quasimomentum states. To measure the dephasing time, we apply an impulse to a spin-polarized gas trapped in the lattice. A force generated via a magnetic field gradient is applied to the atoms along the same direction as the Raman wavevector difference. The strength of the force is tuned to transfer approximately the same momentum to the gas as the Raman excitation.

Sample measurements of the average quasimomentum of the gas for different hold times in a $s = 4$ lattice after the impulse is applied are shown in Fig. 6a for $T/T_F \approx 0.25$ (before the lattice is turned on) and $N \approx 42,000$. We analyze the data in the same way as for Fig. 2 and fit to Eq. 1 to determine a dephasing timescale $\tau$. These data approximately match the condition for the $s = 4$ point in Fig. 5 and the lowest temperature point in Fig. 3.

We simulate the dephasing time by propagating classical trajectories for a thermal distribution of initial quasimomenta subjected to the same impulse as in the experiment. For this simulation, we work in 1D, use 3000 particles, and propagate the position and quasimomentum of each particle according to $dx/dt = \partial H/\partial q$ and $dq/dt = -\partial H/\partial x$ with $H = m\bar{\omega}^2 x^2/2 + 2t[1 - \cos(\pi q/q_B)]$. We weight the quasimomentum of the particles by a FD distribution and determine average quasimomentum for different propagation times. The results of a simulation for $\tilde{\mu} = 4.5t/3$ and $\tilde{T} = 1.3(t/k_B)/3$ (in the lattice) are shown in Fig. 6b. We choose thermodynamic parameters a factor of three times smaller than the corresponding experimental points to account for the three times smaller bandwidth in 1D compared with 3D. With this adjustment, the parameters used for Fig. 6b match the experimental conditions for the lowest temperature point in

Fig. 3b and those in Fig. 6a. As with the experimental data, we determine a dephasing time by fitting the simulated data to Eq. 1.

A summary of the measured and simulated dephasing times is shown in Fig. 6c for different temperatures in an $s = 4$ lattice. The data shown in Fig. 5 will behave similarly to the lowest temperature point. For comparison, the corresponding measured transport lifetimes from Fig. 3b are displayed. As expected, the dephasing time is smaller at higher temperature since a wider range of quasimomenta are present. The agreement between simulated and measured dephasing times for spin-polarized gases indicates that the simulation accurately describes the dephasing dynamics. The simulated dephasing times have much smaller uncertainties than the measurements and therefore are a useful benchmark for estimating the impact of dephasing on our measurements. The simulated dephasing time is at least four times longer than the measured transport lifetime. At high temperature, where the deviation from weak scattering theory is largest, the simulated dephasing time is six times longer than the measured transport lifetime. We conclude that dephasing has a minor impact on our measurements.

**Analog of resistivity**. A complication is that measurements on solids and the DMFT prediction we compare with involve a spatially uniform system, while our measurements are averaged over the inhomogeneous atomic density profile. We must, therefore, correct our measured $\tau_t$ for this average. We use an approach that assumes a null hypothesis: weak binary s-wave scattering for a trapped gas. In that limit, the collision rate per atom is $n_{dwd}\langle v\rangle\sigma$, where $\langle v\rangle$ is the mean relative speed between colliding partners and $\sigma$ is the collision cross-section, and $\rho$ will be independent of temperature (ignoring the effects of Pauli blocking, which are minimal for our measurements). Our construction for resistivity will not correctly account for density or energy-dependent scattering processes, which is precisely the type of phenomena we wish to expose by measuring changes in $\rho$ with temperature.

For any process that generates resistivity through independent two-body scattering, without losing generality, we can write $1/\tau_t = n_s M$, where $M$ is an integral (over momenta) of scattering matrix elements that contribute to the decay of current, and $n_s$ is the density of scatterers. The resistivity is proportional to $m^*/\tau_t$, where $m^*$ is the effective mass. The challenge in determining resistivity for strongly correlated systems is evaluating $M$.

In our case, $n_s = n_\uparrow(\mathbf{r})$, which varies across the gas. Our measured transport lifetime is a weighted average $M\int d^3 r n_\uparrow(\mathbf{r}) n_\downarrow(\mathbf{r})/N_\downarrow = M n_{dwd}$ over the $|\downarrow\rangle$ density profile, which links our measurements with $M$. Here, $n_{dwd}$ is the density-weighted density, and $N_\downarrow$ is the number of atoms in the $|\downarrow\rangle$ state. To compare with DMFT simulations, which involves a uniform system with fixed electron density, we, therefore, divide the measured $\tau_t$ by $1/n_{dwd}$. In addition, since $m^* \propto t$, we absorb a factor of $t$ into a dimensionless $\tau_t/(\hbar/t)$, and define the dimensionless resistivity $\rho = \left(\frac{\tau_t}{\hbar/t} n_{dwd} d^3\right)^{-1}$.

Our analysis also assumes that $\langle v\rangle = \langle |\nabla_\mathbf{q}\, \varepsilon|\rangle$ is independent of temperature. We also assume that the thermally averaged speed of the $|\downarrow\rangle$ component is also fixed with respect to changes in temperature for our determination of $\rho_{MIR}$. Variation in the thermally averaged speeds of the particles is suppressed because $E_F \approx 6t$ and there is a maximum allowed speed in this single-band system. For our experimental parameters, we have used non-interacting thermodynamics to determine that $\langle v\rangle$ is fixed to within 4% and the thermally averaged speed for the $|\downarrow\rangle$ component is fixed to within 1%.

Calculations that allow for variation away from half-filling (unlike the code we employ) predict a density-dependent resistivity. This effect, combined with the inhomogeneous density profile and change in density with temperature, would lead to non $T$-linear behavior for our measurements. To estimate the magnitude of this effect, we use the density dependence $\rho \propto 1/(1 - n)$ predicted in ref. [15] for the $U \to \infty$ and high-temperature limit. This dependence averaged across the $|\uparrow\rangle$ density profile would induce a 10% change in the trap-averaged resistivity at the lowest temperature sampled in Figs. 3 and 4b and a 1% change at the highest temperature. We conclude that this is a minor effect and the $T$-linear scaling predicted by the TRIQS code should apply to the trap-averaged $\rho$.

**Data availability**
The data and computer code that support the findings of this study are available from the corresponding author upon reasonable request.

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

## Acknowledgements

We acknowledge support from the Army Research Office (W911NF-17-1-0171) and National Science Foundation (PHY 15-05468).

## Author contributions

W.X., W.R.M., W.N.M., and B.D. designed the experiment. W.X. and W.N.M. carried out the measurements. W.X., W.N.M., and B.D. analyzed the data. B.D. supervised the project. W.X., W.N.M., and B.D. edited the manuscript. All authors discussed the results and contributed to the manuscript.

## Additional information

**Competing interests:** The authors declare no competing interests.

