## [Peer Review File · Nature Communications]

Reviewers' comments:

Reviewer #1 (Remarks to the Author):

Transport in interacting quantum many body systems in the high-temperature regime is a fundamental problem which is still poorly understood. Because of the absence of phonons and disorder, cold fermionic atoms in optical lattices offer a great platform for exploring this physics, which is the purpose of the present paper. Hence, the general subject and experimental endeavour is both timely and of broad general interest.

The experimental procedure makes sense. An initial perturbation corresponding to a finite current (momentum shift) is generated, and the momentum relaxation is monitored as a function of time, allowing the authors to extract a momentum relaxation time. The experimental setup is a conventional one with an optical lattice subject to harmonic confinement - the latter makes the interpretation of the results significantly more complex and less direct than if a flat trap had been used, because of the inhomogeneous density and especially the fact that the density profile in the trap depends on temperature.

Another point that should be discussed in more depth by the authors is to what extent their procedure keeps the system close to equilibrium - i.e. whether the applied momentum shift can be viewed as a small perturbation or not - is the experiment in the linear response regime, etc.

The main problem of this paper however is that the presentation and analysis of the data as well as the writing of the paper are far from perfect. Here are some of my concerns:

1. Fig.3: This is a key figure displaying the transport lifetime as a function of temperature. Unfortunately, the presentation of the data is, I am afraid to say, quite sloppy. The figure has two vertical axis scales: left and right. The scale on the left has only one number indicated [50] and somewhat strangely spaced tickmarks which make it impossible to decide where the scale starts at. I was interested in knowing what is the lowest value of the ratio $[\tau/(\hbar/t)]$ reached in the experiment but couldn't do it looking at the figure. The right scale also is only a bit more readable, with again tickmarks in strange places (I assume that the scale is logarithmic although not indicated in the caption).

2. Fig4. In this figure, a 'resistivity' is reconstructed by combining the measured transport lifetime with the density of carriers extracted from the thermally averaged density. The dependence of the reconstructed resistivity as a function of temperature is a central result of this paper.

I have several comments:

i) The assumptions that have been made in order to reconstruct the 'resistivity' should be discussed in a more critical manner. In particular, the thermally averaged density is a global quantity over the whole trap, while the actual transport coefficients (the true resistivity) is a local quantity. Hence, one can question this reconstruction procedure, and the authors should devote more space to a convincing discussion of why this reconstruction is a reasonable approximation.

ii) Along those lines, the author could have used theoretical support from the DMFT approach that they use. Indeed, the inhomogeneous density profile *can* be taken into account in DMFT calculations (either using the

LDA approximation or a full real-space DMFT framework, as done by several authors) - contrarily to what is stated on page 8. Such a calculation would have provided guidance in the reconstruction procedure.

iii) It is hard to figure out, based on the presented data (see my remark above about Fig.3), what is the dominant source of temperature dependence of the reconstructed resistivity: does the reported linear-T dependence stem mostly from the T-dependence of the transport lifetime, mostly from the T-dependence of the density, or from the interplay of both? In particular, a figure (or inset to Fig.4) displaying the temperature dependence of n_{dwd} would be welcome.

iv) In the same spirit, a measurement of the compressibility (easily reachable by varying the trap confinement) would have been useful. In the very high-T regime, it is well established theoretically that the T-linear behaviour of the resistivity in the Hubbard model is entirely due to the behaviour of the compressibility and not of the transport relaxation time (or diffusion constant).

These are the main points of concern regarding the results of this article. There are other points of lesser importance, which can be easily addressed and which would help putting the results in context :

a. There is some confusion in terminology between 'bad' and 'strange' metals. Admittedly, the condensed-matter physics literature is not always clear about this. In my opinion, the term 'bad metal' should be used only in the regime where the resistivity is larger than the Mott-Ioffe-Regel limit (this is what the authors of Ref.4 had in mind when introducing this terminology). In contrast 'strange' metallic transport should be used when the resistivity is *less* than the MIR value, but cannot be explained by existing theories (hence 'strange') - in particular does not follow Fermi liquid behaviour. This is the case of cuprates *close to optimal doping* for example (not in the underdoped regime, in which bad metallic behaviour applies).

b. In this context, comparing the transport lifetime to \hbar/t is crucial - hence my remark about Fig.3. Whenever this ratio is smaller or comparable to unity, we have bad metallic behaviour.

c. The introduction could be written in a better manner to clarify the context. In particular, important references are missing, e.g.:

- A detailed study of the crossover between 'bad metal' transport at hi-T and Fermi liquid at low-T was provided (in the DMFT context) by Deng et al. Phys Rev Lett 110, 086401 (2013). In this work, it was shown that, within the DMFT approximation, 'strange' metal behaviour below the MIR applies but that quasiparticles are present in this regime while they indeed disappear at the MIR value).

- The relevance of cold atoms experiments in probing hi-T transport was advertised in Ref.36, as which also provides a detailed understanding of the hi-T bad metal regime of the Hubbard model.

- Linear-T behaviour of the resistivity was discussed in the context of SYK models, first in Parcollet et al.

PRB 59, 5341 (1999) and very recently by several authors.

- The role of phonons in the context of resistivity saturation, which is alluded to by the authors in the introduction,

has been discussed in recent theoretical work by E. Berg et al., see e.g. arXiv:1705.07895, NPJ QUantum Materials 2, 7 (2016), Phys Rev B 93, 075109 (2016).

d. Some quantities are introduced in the text without explanation, e.g. the spectral function $A(\omega)$ on page 10.

e- In relation to the spectral function, some incorrect statements are also made:

- ``At low-T, $A(\omega)$ consists of a single peak at the Fermi level that is broadened by interactions': this statement would be correct for the *momentum-resolved* spectral function as measured in ARPES.

In contrast, the authors discuss the momentum-integrated $A(\omega)$: it always has a finite width due to the k -integration, which corresponds to the bandwidth of quasiparticles, not to their lifetime.

- ``This change in $A(\omega)$ gives rise to a resistivity that increases more rapidly with T than the Fermi liquid prediction': actually in DMFT (see Deng et al above) this crossover leads to a resistivity increasing

less rapidly than what Fermi liquid behaviour would yield, when extrapolated in this regime.

To summarize: this is an interesting article, addressing a fundamental question in a timely manner. It should be acknowledged that a first version was submitted (and appeared as arXiv:1606.06669) well before the article of Bakr et al. (arXiv:1802.09456) addressing a similar question. A such, it should be, in my opinion, given a chance for publication in a journal such as Nature Communications.

However, I cannot recommend publication of the manuscript in the current state.

I would recommend that the authors improve their manuscript taken into account the points above and consider resubmitting.

Reviewer #2 (Remarks to the Author):

The manuscript by Xu and coworkers reports an experimental investigation of the relaxation dynamics of ultracold neutral fermions in lattices. Their setup simulates the resistivity of the Fermi-Hubbard model in a wide range of temperatures, not accessible in real solids, and in the absence of the typical complications that are present in solids (e.g. phonons and disorder, which however might be introduced later in a controllable fashion).

The authors develop a novel technique to excite the motion of one of the spin components in the system and to measure a characteristic transport lifetime. From the lifetime they extract the analogue

of the resistivity of a real solid, and study how the resistivity changes with the temperature.

The system investigated by Xu and coworkers is in a totally different temperatures regime compared to real metals, since T is comparable to T_F or can get even larger than T_F . Despite that difference, the authors discuss how the basic phenomenon observed in this work might be helpful to understand some aspects of the so-called "bad metals".

They indeed find that the resistivity increases almost linearly with T , similarly to what happens in bad metals, which show a similar deviation from the usual T^2 behavior of the Fermi liquid theory. The authors discuss another characteristic feature of bad metals, the absence of saturation of the resistivity at the so-called Motte-Ioffe-Regel (MIR) limit. Although they can reach a regime in which the resistivity approaches the MIR curve, they have no proof of a violation of the limit.

The study reported in the manuscript looks quite interesting, as it tries to connect similar fundamental transport phenomena observed in two different physics sectors. The efforts to develop a transport experiment and to explore how the resistivity depends on temperature must be recognized as an important step forward in the field. The authors correctly note that there is a concurrent study by another group (ref. 34) which makes similar observations on a slightly different, 2D system.

In my opinion, the results of this work are potentially of general interest; the measurements are of high quality and the analysis is convincing. However, I cannot yet make a recommendation towards publication in Nature Communication, since there are a couple of aspects of the discussion which do not convince me and I think require some clarifications.

1) My first point is about the MIR limit. In the abstract, the authors state that they "observe the onset of two behaviors- incoherent transport and the approach to the MIR limit – associated to bad metals". The observed resistivity approaches the MIR limit without any doubt, as shown in Fig.4, but I would not say that the data demonstrates any absence of a saturation at ρ_{MIR} . In particular, in Fig.4b there is only one data point at large T that deviates from the theory line and approaches the MIR limit. I would then conclude that the following sentence is unsubstantiated: "The onset of another characteristic of bad metals is also evident in Fig.4: absence of saturation as the MIR limit is approached". I would instead agree with the later sentence "While we measure a steady increase of resistivity with temperature and interaction strength, a violation of the MIR limit is not evident." I think that the authors should explain more clearly how they conclude that a saturation is not present. Is their conclusion based on the experimental data of the resistivity vs the interaction strength, or vs the temperature? Or is it based on the DMFT predictions?

2) My second point regards the discussion in Section III about the spectral function A . I find the sentence "The transfer of spectral weight at higher T to peaks centered at $\pm U/2$... is indicative of the cross-over to a bad metal and the state becoming localized." a bit obscure. First, in Fig4.b or in the text I could not find the value of U at which A was calculated. Second, it might be not clear to all readers that the appearance of the side peaks is the indication of a tendency to localization of the particles. I therefore suggest explaining it more clearly, perhaps also moving ref.33 into that sentence. Third, I'm not fully convinced that the reasoning of ref. 33 applies also for the high- T system studied by the authors. To my understanding, the localization mentioned in ref.33 is that of a Mott insulator, which might be reached in a low- T system but I guess it would melt at the high temperatures explored in this work.

Here are a few minor suggestions for the presentation.

1) Please define the acronym MIR in the introduction.

2) In the insets of Fig.2b, I found difficult to understand the collisional processes at first glance, since there is little contrast between the colored particles and the colored lattice. Please consider using a monochromatic lattice.

3) In Fig.3, please consider labelling the ticks at 1 and 10 of the vertical scale on the right, instead of 1.1 and 11.5.

Reviewer #3 (Remarks to the Author):

This is a referee report on the manuscript "Bad-Metal Relaxation Dynamics in a Fermi Lattice Gas" by W. Xu et al.

In this paper, the Authors present experimental observations of unusual temperature dependence in the decay of transport in an ultracold atomic system, which is presented as evidence of "bad metal" behaviour in this Fermionic system.

In summary, the Authors prepared a spin-polarised sample, and transferred some of the atoms (~30%) to a different internal state with a two-photon process that gives them a momentum kick. By monitoring the time evolution of these kicked atoms, they are able to extract an effective resistivity of the system, and study it as a function of the temperature and the (contact) interactions of the system.

The main message of their experiments is that they access a parameter regime where the resistivity decreases upon increasing temperature -- a behaviour incompatible with Landau's Fermi liquid theory, which thus points to strong correlations in the system. This is further supported by the disagreement of the observations with calculations from a weakly-interacting model ("Fermi golden rule"). Together with the DMFT calculations (Supp. Note 3), I consider that overall the authors have made a good case to support that their observed resistivity presents a linear dependence on temperature, which is characteristic of bad metals, for a good part of the accessed parameter regime. On the other hand, the considerable errors bars in their data at increasing temperatures [Fig. 4] make it harder to make a strong case for their system to approach the Mott-Ioffe-Regel limit; perhaps the very different nature of the lattice here, without phonons, as compared with solid state samples, can explain this discrepancy.

The paper is written in a clear way, and presents with sufficient detail the main points of the experiments performed in the main text, with details on methods conveniently described in the Methods section and Supplementary Sections. The methods employed appear suitable for the problem at hand.

This work is a clear demonstration of the emergence of ultracold atomic systems as quantum simulators, enabling access to parameter regimes not accessible in conventional solid state systems, with microscopic understanding of the physics of the system. Because of this, I consider this manuscript satisfies the requirements of rigour and expected impact in the field for publication in Nature Communications.

A few minor points that the Authors should consider before publication are:

- In Fig. 3, add further axis marks on the y-axis, to make clear when the incoherent regime is expected to appear, similarly to Fig. 5.
- It is unclear the extent of the "reduction of quasiparticle weight" (that "explains the qualitative failure of the weak scattering calculation") from the DMFT calculations in Fig. 4: given that all plots of $A(\omega)$ are supposed to be on the same scale, it is not apparent from the figure that any peaks loose strength on increasing U . Could the authors elaborate more on this point?

We thank the reviewers for their thoughtful reports, and we appreciate their consensus that this work is worthy of publication with appropriate clarifications. In the response that follows, we conclusively address all of the referee questions, concerns, and comments. We hope that the manuscript can now move swiftly toward publication.

Our responses (in black typeface) to the individual referee comments (in red typeface) follow.

Reviewer #1 (Remarks to the Author):

Transport in interacting quantum many body systems in the high-temperature regime is a fundamental problem which is still poorly understood. Because of the absence of phonons and disorder, cold fermionic atoms in optical lattices offer a great platform for exploring this physics, which is the purpose of the present paper. Hence, the general subject and experimental endeavour is both timely and of broad general interest.

The experimental procedure makes sense. An initial perturbation corresponding to a finite current (momentum shift) is generated, and the momentum relaxation is monitored as a function of time, allowing the authors to extract a momentum relaxation time. The experimental setup is a conventional one with an optical lattice subject to harmonic confinement - the latter makes the interpretation of the results significantly more complex and less direct than if a flat trap had been used, because of the inhomogeneous density and especially the fact that the density profile in the trap depends on temperature.

Another point that should be discussed in more depth by the authors is to what extent their procedure keeps the system close to equilibrium - i.e. whether the applied momentum shift can be viewed as a small perturbation or not - is the experiment in the linear response regime, etc.

We are not aware of a universally agreed upon method for characterizing closeness to equilibrium. Therefore, in the manuscript, we provided enough information for the reader to understand the size of the excitation relative to other characteristic scales. We explain all the excitation parameters (e.g, the size of the excitation in quasimomentum and the number of atoms involved). And, in an attempt to characterize the strength of the excitation and deviation from equilibrium, we explained that the excitation increases the total energy of the gas by less than 10%.

The issue of non-linear response is important. We explained in the manuscript that the linear model fits the data well by giving a range of adjusted R^2 . However, interpreting R^2 values is complicated for non-linear regression. Therefore, we have carried out an additional analysis of the data (which is included in the Supplementary Information) to check whether non-linear response is present. Our analysis and solution to the Boltzmann equation assumes linear damping, i.e., the equivalent of ohmic response. If

non-linearity is present, the residuals for the fits used to determine τ_t will show systematic shifts (from zero) that vary with hold-time. Sample residuals are shown across the full range of data in Supplementary Figures 7 and 8. The uniform scatter around zero and absence of systematic shifts with t_{hold} indicate that non-linear effects are minor.

The main problem of this paper however is that the presentation and analysis of the data as well as the writing of the paper are far from perfect. Here are some of my concerns:

1. Fig.3: This is a key figure displaying the transport lifetime as a function of temperature. Unfortunately, the presentation of the data is, I am afraid to say, quite sloppy. The figure has two vertical axis scales: left and right. The scale on the left has only one number indicated [50] and somewhat strangely spaced tickmarks which make it impossible to decide where the scale starts at. I was interested in knowing what is the lowest value of the ratio $[\tau_t/(\hbar/t)]$ reached in the experiment but couldn't do it looking at the figure. The right scale also is only a bit more readable, with again tickmarks in strange places (I assume that the scale is logarithmic although not indicated in the caption).

We agree with the reviewer that the axes in this figure are poorly labeled. We made an error late in the editing process. This issue has been fixed.

2. Fig4. In this figure, a 'resistivity' is reconstructed by combining the measured transport lifetime with the density of carriers extracted from the thermally averaged density. The dependence of the reconstructed resistivity as a function of temperature is a central result of this paper.

I have several comments:

i) The assumptions that have been made in order to reconstruct the 'resistivity' should be discussed in a more critical manner. In particular, the thermally averaged density is a global quantity over the whole trap, while the actual transport coefficients (the true resistivity) is a local quantity. Hence, one can question this reconstruction procedure, and the authors should devote more space to a convincing discussion of why this reconstruction is a reasonable approximation.

The observables we infer are necessarily thermodynamic averages over the density profile.

The way we incorporate this average when constructing a resistivity is not an approximation, but rather obeys the null hypothesis of relaxation driven by weak, binary s-wave scattering described by an energy and density independent cross-section. Inherent in this approach is a local density approximation, i.e., that plane-wave

scattering occurs. The literature explaining this approach and demonstrations of its success for weakly interacting trapped Fermi gases are referenced in the paper. For weak scattering, the collision rate per quasiparticle is $\Gamma = n_{dwd}\sigma v_{rel}$, where σ is the collision cross-section (that must be corrected for Pauli blocking at low temperatures) and v_{rel} is the (thermodynamically averaged) mean relative speed between colliding partners. When a weak scattering picture is appropriate, all relaxation measurements occur with a timescale inversely proportional to Γ .

As we discuss in the manuscript, the resistivity is constructed such that it is independent of temperature for the null hypothesis. The temperature dependence we observe is thus a signature of a violation of weak scattering.

We have been more explicit and direct in the manuscript about using a null hypothesis and the limitations of averaging over the density profile and the local density approximation. For example, we explain that the effect of a density-dependent transport coefficient cannot be captured by this approach, and that this type of effect is precisely what we are trying to expose.

In the manuscript, we also now address another assumption involving the density profile that is inherent in comparing the inferred resistivity to predictions from the DMFT code we use. The version of the TRIQS codebase we employed only handles fixed density at half filling. Since a more sophisticated DMFT approach predicts a density-dependent resistivity, it is possible that the T -linear scaling (if correct in a uniform system) is altered by averaging over the density profile. To estimate the deviation from T -linear scaling, we use a result from PRB **94**, 235115 (2016). The authors of that work show via a series expansion method that in the high-temperature and $U \rightarrow \infty$ limit, $\rho \propto \frac{1}{1-n}T$. Furthermore, the authors demonstrate that the series expansion is an excellent approximation to NRG-DMFT calculations in this regime.

For the range of temperatures we sample in Fig. 3 and 4b, we estimate the impact of this density dependence on T -linear scaling by averaging $\frac{1}{1-n}$ over the density profile. At the lowest temperature, this density dependence would change the predicted resistivity by 10%, and at the highest temperature by 1%. We conclude that the density dependence of resistivity predicted by DMFT is sufficiently weak such that T -linear scaling should be preserved for the trap-averaged resistivity.

ii) Along those lines, the author could have used theoretical support from the DMFT approach that they use. Indeed, the inhomogeneous density profile *can* be taken into account in DMFT calculations (either using the LDA approximation or a full real-space DMFT framework, as done by several authors) - contrarily to what is stated on page 8. Such a calculation would have provided guidance in the reconstruction procedure.

We thank the referee for bringing this work to our attention. Our misstatement

regarding the ability of DMFT to take into account an inhomogeneous density profile has been corrected. In this section of the text, we have also clarified our derivation of resistivity as based on a null hypothesis.

A limitation of the DMFT code we use for this work is that it can only handle half-filling. Directly taking into account the inhomogeneous density profile in DMFT is beyond the scope of this work.

iii) It is hard to figure out, based on the presented data (see my remark above about Fig.3), what is the dominant source of temperature dependence of the reconstructed resistivity: does the reported linear-T dependence stem mostly from the T-dependence of the transport lifetime, mostly from the T-dependence of the density, or from the interplay of both? In particular, a figure (or inset to Fig.4) displaying the temperature dependence of n_{dwd} would be welcome.

The strong temperature dependence of the inferred resistivity is a result of the transport lifetime changing very little over the temperature range we sample, while the density-weighted density changes significantly. That is: most of the change in inferred resistivity is due to the variation in n_{dwd} with temperature. As we explained in the manuscript, for the weak-scattering null hypothesis, the change in inverse transport lifetime should be compensated by n_{dwd} (since v_{rel} is fixed), thereby resulting in a temperature-independent resistivity. The T -linear scaling we observe is a signature that this weak scattering behavior is violated.

We note the linear dependence of inverse relaxation time on n_{dwd} has been observed for weakly interacting trapped Fermi gases. In fact, that dependence was used to make the first measurement of the s -wave collision cross section and scattering length for ultracold ^{40}K atoms.

We have included a plot of n_{dwd} in the Supplementary Information.

iv) In the same spirit, a measurement of the compressibility (easily reachable by varying the trap confinement) would have been useful. In the very high-T regime, it is well established theoretically that the T-linear behaviour of the resistivity in the Hubbard model is entirely due to the behaviour of the compressibility and not of the transport relaxation time (or diffusion constant).

Two issues complicate inferring compressibility by measuring changes in the size of the gas as the trap frequency is varied. First, a significant fraction of the atoms is located in the low-density, highly compressible region at the edge of the gas in 3D. Second, the images we take of the 3D gas are column integrated, which masks changes in the rms size of the central, high-density region. For these reasons, it is not feasible to infer compressibility in an unambiguous way (and without extensive modeling) by changing the trap frequency.

An alternative method that is less complicated to interpret is to measure core compressibility, which is beyond the scope of the present work.

These are the main points of concern regarding the results of this article. There are other points of lesser importance, which can be easily addressed and which would help putting the results in context :

a. There is some confusion in terminology between 'bad' and 'strange' metals. Admittedly, the condensed-matter physics literature is not always clear about this. In my opinion, the term 'bad metal' should be used only in the regime where the resistivity is larger than the Mott-Ioffe-Regel limit (this is what the authors of Ref.4 had in mind when introducing this terminology). In contrast 'strange' metallic transport should be used when the resistivity is *less* than the MIR value, but cannot be explained by existing theories (hence 'strange') - in particular does not follow Fermi liquid behaviour. This is the case of cuprates *close to optimal doping* for example (not in the underdoped regime, in which bad metallic behaviour applies).

b. In this context, comparing the transport lifetime to \hbar/t is crucial - hence my remark about Fig.3 Whenever this ratio is smaller or comparable to unity, we have bad metallic behaviour.

We agree with the referee that the terms "bad metal" and "strange metal" are not used consistently in the literature. We have been careful to define precisely what we mean by "bad metal" in the paper.

We agree with the referee that the transport lifetime approaching \hbar/t is an important part of the evidence for bad metallic behavior. The commensurability between these timescales is now more clearly evident in Fig. 3.

c. The introduction could be written in a better manner to clarify the context. In particular, important references are missing, e.g.:

- A detailed study of the crossover between 'bad metal' transport at hi-T and Fermi liquid at low-T was provided (in the DMFT context) by Deng et al. Phys Rev Lett 110, 086401 (2013). In this work, it was shown that, within the DMFT approximation, 'strange' metal behaviour below the MIR applies but that quasiparticles are present in this regime while they indeed disappear at the MIR value).

- The relevance of cold atoms experiments in probing hi-T transport was advertised in Ref.36, as which also provides a detailed understanding of the hi-T bad metal regime of the Hubbard model.

- Linear-T behaviour of the resistivity was discussed in the context of SYK models, first in

Parcollet et al. PRB 59, 5341 (1999) and very recently by several authors.

- The role of phonons in the context of resistivity saturation, which is alluded to by the authors in the introduction, has been discussed in recent theoretical work by E. Berg et al., see e.g. arXiv:1705.07895, NPJ QUantum Materials 2, 7 (2016), Phys Rev B 93, 075109 (2016).

We have added these references to the paper.

d. Some quantities are introduced in the text without explanation, e.g. the spectral function $A(\omega)$ on page 10.

We have provided a standard reference for the local spectral function and provided more information in the Supplementary Information.

e- In relation to the spectral function, some incorrect statements are also made:

- "At low-T, $A(\omega)$ consists of a single peak at the Fermi level that is broadened by interactions": this statement would be correct for the *momentum-resolved* spectral function as measured in ARPES. In contrast, the authors discuss the momentum-integrated $A(\omega)$: it always has a finite width due to the k-integration, which corresponds to the bandwidth of quasiparticles, not to their lifetime.

We agree with the referee, and this statement has been fixed.

- "This change in $A(\omega)$ gives rise to a resistivity that increases more rapidly with T than the Fermi liquid prediction": actually in DMFT (see Deng et al above) this crossover leads to a resistivity increasing *less* rapidly than what Fermi liquid behaviour would yield, when extrapolated in this regime.

We have clarified this point in the text.

To summarize: this is an interesting article, addressing a fundamental question in a timely manner. It should be acknowledged that a first version was submitted (and appeared as arXiv:1606.06669) well before the article of Bakr et al. (arXiv:1802.09456) addressing a similar question. As such, it should be, in my opinion, given a chance for publication in a journal such as Nature Communications. However, I cannot recommend publication of the manuscript in the current state. I would recommend that the authors improve their manuscript taken into account the points above and consider resubmitting.

Reviewer #2 (Remarks to the Author):

The manuscript by Xu and coworkers reports an experimental investigation of the relaxation dynamics of ultracold neutral fermions in lattices. Their setup simulates the resistivity of the Fermi-Hubbard model in a wide range of temperatures, not accessible in real solids, and in the absence of the typical complications that are present in solids (e.g. phonons and disorder, which however might be introduced later in a controllable fashion).

The authors develop a novel technique to excite the motion of one of the spin components in the system and to measure a characteristic transport lifetime. From the lifetime they extract the analogue of the resistivity of a real solid, and study how the resistivity changes with the temperature.

The system investigated by Xu and coworkers is in a totally different temperatures regime compared to real metals, since T is comparable to T_F or can get even larger than T_F . Despite that difference, the authors discuss how the basic phenomenon observed in this work might be helpful to understand some aspects of the so-called “bad metals”.

They indeed find that the resistivity increases almost linearly with T , similarly to what happens in bad metals, which show a similar deviation from the usual T^2 behavior of the Fermi liquid theory. The authors discuss another characteristic feature of bad metals, the absence of saturation of the resistivity at the so-called Motte-Ioffe-Regel (MIR) limit. Although they can reach a regime in which the resistivity approaches the MIR curve, they have no proof of a violation of the limit.

The study reported in the manuscript looks quite interesting, as it tries to connect similar fundamental transport phenomena observed in two different physics sectors. The efforts to develop a transport experiment and to explore how the resistivity depends on temperature must be recognized as an important step forward in the field. The authors correctly note that there is a concurrent study by another group (ref. 34) which makes similar observations on a slightly different, 2D system.

In my opinion, the results of this work are potentially of general interest; the measurements are of high quality and the analysis is convincing. However, I cannot yet make a recommendation towards publication in Nature Communication, since there are a couple of aspects of the discussion which do not convince me and I think require some clarifications.

1) My first point is about the MIR limit. In the abstract, the authors state that they “observe the onset of two behaviors- incoherent transport and the approach to the MIR limit – associated to bad metals”. The observed resistivity approaches the MIR limit without any doubt, as shown in Fig.4, but I would not say that the data demonstrates

any absence of a saturation at ρ_{MIR} . In particular, in Fig.4b there is only one data point at large T that deviates from the theory line and approaches the MIR limit. I would then conclude that the following sentence is unsubstantiated: “The onset of another characteristic of bad metals is also evident in Fig.4: absence of saturation as the MIR limit is approached”. I would instead agree with the later sentence “While we measure a steady increase of resistivity with temperature and interaction strength, a violation of the MIR limit is not evident.” I think that the authors should explain more clearly how they conclude that a saturation is not present. Is their conclusion based on the experimental data of the resistivity vs the interaction strength, or vs the temperature? Or is it based on the DMFT predictions?

We apologize for the confusion caused by our imprecise language. It is correct that we do not observe the MIR limit to be violated. By “absence of saturation,” we mean that the inferred ratio ρ/ρ_{MIR} (shown in the inset to Fig. 4) steadily approaches the MIR limit (i.e., the slope of that ratio vs. temperature does not change with temperature).

We have removed “absence of saturation” from the text and clarified this point.

2) My second point regards the discussion in Section III about the spectral function A . I find the sentence “The transfer of spectral weight at higher T to peaks centered at $\pm U/2$... is indicative of the cross-over to a bad metal and the state becoming localized.” a bit obscure. First, in Fig4.b or in the text I could not find the value of U at which A was calculated. Second, it might be not clear to all readers that the appearance of the side peaks is the indication of a tendency to localization of the particles. I therefore suggest explaining it more clearly, perhaps also moving ref.33 into that sentence. Third, I’m not fully convinced that the reasoning of ref. 33 applies also for the high-T system studied by the authors. To my understanding, the localization mentioned in ref.33 is that of a Mott insulator, which might be reached in a low-T system but I guess it would melt at the high temperatures explored in this work.

We have added the value of U at which A was calculated (which matches the data) to the figure caption.

Theory and numerical work by various authors has determined that the effect of finite temperature is to shift spectral weight into the Mott lobes, stabilize the Mott insulating regime, and increase localization behavior and resistivity near the onset of a Mott transition. This was observed by, for example, Pakhira et al (Ref. 16 in the text). Within DMFT, this framework is one way of understanding the origins of bad metal behavior. We have added this elaboration to the text.

Here are a few minor suggestions for the presentation.

- 1) Please define the acronym MIR in the introduction.
- 2) In the insets of Fig.2b, I found difficult to understand the collisional processes at first

glance, since there is little contrast between the colored particles and the colored lattice. Please consider using a monochromatic lattice.
3) In Fig.3, please consider labelling the ticks at 1 and 10 of the vertical scale on the right, instead of 1.1 and 11.5.

We thank the referee for these comments, which we have incorporated.

Reviewer #3 (Remarks to the Author):

This is a referee report on the manuscript "Bad-Metal Relaxation Dynamics in a Fermi Lattice Gas" by W. Xu et al.

In this paper, the Authors present experimental observations of unusual temperature dependence in the decay of transport in an ultracold atomic system, which is presented as evidence of "bad metal" behaviour in this Fermionic system.

In summary, the Authors prepared a spin-polarised sample, and transferred some of the atoms (~30%) to a different internal state with a two-photon process that gives them a momentum kick. By monitoring the time evolution of these kicked atoms, they are able to extract an effective resistivity of the system, and study it as a function of the temperature and the (contact) interactions of the system.

The main message of their experiments is that they access a parameter regime where the resistivity decreases upon increasing temperature -- a behaviour incompatible with Landau's Fermi liquid theory, which thus points to strong correlations in the system. This is further supported by the disagreement of the observations with calculations from a weakly-interacting model ("Fermi golden rule"). Together with the DMFT calculations (Supp. Note 3), I consider that overall the authors have made a good case to support that their observed resistivity presents a linear dependence on temperature, which is characteristic of bad metals, for a good part of the accessed parameter regime. On the other hand, the considerable errors bars in their data at increasing temperatures [Fig. 4] make it harder to make a strong case for their system to approach the Mott-Ioffe-Regel limit; perhaps the very different nature of the lattice here, without phonons, as compared with solid state samples, can explain this discrepancy.

The paper is written in a clear way, and presents with sufficient detail the main points of the experiments performed in the main text, with details on methods conveniently described in the Methods section and Supplementary Sections. The methods employed appear suitable for the problem at hand.

This work is a clear demonstration of the emergence of ultracold atomic systems as quantum simulators, enabling access to parameter regimes not accessible in conventional solid state systems, with microscopic understanding of the physics of the system. Because of this, I consider this manuscript satisfies the requirements of rigour

and expected impact in the field for publication in Nature Communications.

A few minor points that the Authors should consider before publication are:

- In Fig. 3, add further axis marks on the y-axis, to make clear when the incoherent regime is expected to appear, similarly to Fig. 5.

We have made this change.

- It is unclear the extent of the "reduction of quasiparticle weight" (that "explains the qualitative failure of the weak scattering calculation") from the DMFT calculations in Fig. 4: given that all plots of $A(\omega)$ are supposed to be on the same scale, it is not apparent from the figure that any peaks lose strength on increasing U . Could the authors elaborate more on this point?

We apologize that this point was not clear in the text. The relevant effect is the change in the shape of the spectral function and the redistribution of spectral weight from the vicinity of $\omega/W = 0$ (i.e., the Fermi energy) to higher frequencies. We have clarified this point in the text.

REVIEWERS' COMMENTS:

Reviewer #1 (Remarks to the Author):

I am satisfied by the detailed reply to my previous comments, and by the changes made to the paper which have improved it significantly. In view of the general interest and relevance of the experimental results presented by the authors, I am now happy to recommend publication in Nature Communications without reservations.

Reviewer #2 (Remarks to the Author):

The authors have convincingly improved the manuscript, taking into account all my questions and suggestions. I can now recommend acceptance for publication in Nature Communications.

Reviewer #3 (Remarks to the Author):

This is a referee report on the revised version of the manuscript "Bad-Metal Relaxation Dynamics in a Fermi Lattice Gas" by W. Xu et al.

The authors have improved their manuscript and supplementary material, clarifying the main issues that were raised by the referees. In particular, that have added two new sections as supplementary information where the effect of temperature on the (inhomogeneous) density profile through the density-weighted density is made clearer, and how this impacts the interpretation of the results from the viewpoint of linear response is more accessible to broad readership.

The authors have also improved the presentation of experimental results in Fig. 3, as well as the interpretation of the spectral density $A(\omega)$ in Fig. 4.

Taking this into account, together with the extensive response in their resubmission letter, I consider the paper is now suitable for publication in Nature Communications.